

# Finite temperature spin diffusion in the Hubbard model in the strong coupling limit

Oleksandr Gamayun[1,2⋆], Arthur Hutsalyuk[3], Balázs Pozsgay[3] and Mikhail B. Zvonarev[4]

**1** London Institute for Mathematical Sciences, Royal Institution,
21 Albemarle St, London W1S 4BS, UK
**2** Faculty of Physics, University of Warsaw, ul. Pasteura 5, 02-093 Warsaw, Poland
**3** MTA-ELTE "Momentum" Integrable Quantum Dynamics Research Group,
Department of Theoretical Physics, Eötvös Loránd University,
Pázmány Péter stny. 1A, 1117 Budapest, Hungary
**4** Université Paris-Saclay, CNRS, LPTMS, 91405, Orsay, France

⋆ og@lims.ac.uk

## Abstract

We investigate finite temperature spin transport in one spatial dimension by considering the spin-spin correlation function of the Hubbard model in the limiting case of infinitely strong repulsion. We find that in the absence of a magnetic field the transport is diffusive, and derive the spin diffusion constant. Our approach is based on asymptotic analysis of a Fredholm determinant representation. The obtained results are in agreement with Generalized Hydrodynamics approach.

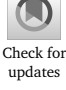

# 1  Introduction

Quantum transport in the integrable systems attracts ever increasing attention of the physics community [1]. Distinctive features of these systems – a completely elastic and factorized (two-body reducible) scattering, and a presence of an infinite number of conservation laws – combined with basic principles of hydrodynamics resulted in the formulation of the Generalized Hydrodynamics (GHD) [2, 3]. In less than a decade, the GHD evolved into a matured field of research [4]. It offers a systematic treatment of ballistic transport in integrable models [2, 3, 5, 6], an example being calculation of finite temperature Drude weights [5, 7, 8], until then requiring case-by-case approach [9–11]. The analysis of non-ballistic (that is, diffusive) transport, along with computation of diffusion constants, can also be tackled within the GHD framework, with the use of thermodynamic form factors or hydrodynamic projections [12–15]. This includes treating anomalous diffusion found in systems possessing special nonabelian symmetries, reviewed in Ref. [16]. The use of the GHD for systems quenched far from equilibrium is also possible [17].

The GHD is an asymptotically exact theory aimed at capturing the dynamics at large distances and past a long-time evolution. It is desirable to complement its findings with first-principle microscopic calculations, making use of exact solvability of integrable models. This has been done for current mean values [18–20] and for the Drude weights in some cases [11, 21]. As a general rule, however, diffusion constants have not been extracted from exact solutions of many-body integrable quantum systems so far. The reason is that the structure of the exact (Bethe-ansatz) wave functions is complicated, and getting closed-form tractable expressions for dynamical correlation functions requires extremely involved resummation procedures for the matrix elements [22]. For example, there exist expressions for dynamical correlation functions in the Heisenberg spin-1/2 chain [23–30], but its finite temperature diffusion constants have not yet been found in this manner.

A way to proceed further is to shift the focus to the models having particularly simple Bethe Ansatz solution and yet non-trivial interparticle interactions. In the case of classical cellular automata such selected models include the Rule54 model [31], box-ball systems [32, 33], and a particle hopping model with two-color excitations [34–36]. A number of physical quantities (relevant also to transport of conserved quantities) were derived exactly in these models, starting from the fundamental equations of motion. In the case of quantum spin chains promising candidates are their large coupling limits. High temperature transport of the Heisenberg spin chain in the large anisotropy limit is described by the folded XXZ model [37–39]. Another useful model with infinite dimensional local spaces is the infinite coupling limit of the *q*-boson model (the phase model), whose real-time dynamics is tractable within the Bethe ansatz approach [40–42]. Finally, predictions of GHD for the one-dimensional Hubbard model can be tested in the limiting case of infinitely strong repulsion. Studying the spin transport and deriving the exact analytic formulas for diffusion constant in that limiting case free of any assumptions is the subject of our work.

The Hubbard model is one of the basic models in physics. It is exactly solvable in one spatial dimension by the Bethe ansatz [22, 43, 44], providing full information about the many-body excitation spectrum and collective phenomena, such as spin-charge separation. The integrability of the one-dimensional Hubbard model is proven within the Yang-Baxter framework using the $R$-matrix of Shastry [45, 46]. The exact solution involves an interplay of fermion and spin degrees of freedom, and is consequently more complicated than those for some other well known integrable models, including the Heisenberg spin-1/2 chain, and the $q$-boson model. Tractable analytic results for correlation functions at and far from equilibrium exist for rather particular observables, and initial conditions [47]. The GHD solution of the Hubbard model has been worked out in Refs. [48–51], and is not yet complemented by the use of the exact solution for dynamical correlations.

The Hubbard model in the limiting case of infinitely strong repulsion, known as the $t-0$ model or the restricted hopping model, has been discussed extensively in the literature [44, 52] (as well as its bosonic counterpart, the Maassarani-Mathieu spin chain, also known as the $SU(3)$ XX model [53–55]). The spectral functions were studied in [56–58]. The coordinate Bethe Ansatz solution of the model has been used to calculate finite temperature correlation functions in Ref. [59]. An alternative representation for its solution, further elaborated in the works [60, 61], has provided grounds for the investigation of real time dynamics in Refs. [62, 63] followed by [64, 65].

In the infinite coupling limit the double occupancies of the Hubbard model are forbidden, they are projected out from the Hilbert space. As an effect the $t-0$ model has a three dimensional local Hilbert space: the local basis states are the vacuum, and the two different single particle states, corresponding to the original Hubbard fermions with the two different spin orientations. The special dynamical properties of the $t-0$ model follow from the projection procedure and the allowed hopping terms of the original Hamiltonian: One can easily show that the spatial ordering of the spins of the electrons is not changed during time evolution, and the time evolution of the positions of the electrons does not depend on the spin configuration. These dynamical phenomena were called "single-file property" and "charge inertness" in [66]. These properties underly the exact solvability of real time dynamics the model.

In this work we focus on the finite temperature spin-spin correlation function in the $t-0$ model. We start with the derivation of the exact results using spin-charge separations. The correlation function can be presented as an integral of the Fredholm determinants for which we perform the asymptotic analysis using a heuristic method of the effective form factors [67–69]. Performing saddle point analysis of the obtained expressions we observed that depending on the initial profile the correlation function in question contains both the ballistic and the diffusive parts. From these expressions we extract value for the Drude weight and the diffusion constant correspondingly. Appendices contain all necessary technical derivations. Our results agree with those obtained from GHD, which we demonstrate in Sec. (5). For the infinite temperature the value of the diffusion constant agrees with the one given in [70]. To the best of our knowledge this is the first time that finite temperature spin diffusion was treated in an interacting lattice model via the exact formulas valid at thermodynamic limit at all times and distances. Also, it is a first quantum mechanical extension of the results of [34–36] regarding models with the "single-file" property.

## 2 Model and spin-charge separation

In this section we introduce the model and a basis separating spin and charge excitations. This basis is well suited to calculate dynamical correlation functions of the model exactly, which we do in section 3.

We consider the Hubbard model describing interacting spin-1/2 fermions on a one-dimensional lattice. The Hamiltonian reads

$$H = -\sum_{\substack{j=-\infty \\ \alpha=\uparrow,\downarrow}}^{\infty} (\psi_{j\alpha}^\dagger \psi_{j+1\alpha} + \psi_{j+1\alpha}^\dagger \psi_{j\alpha}) - hN + 2BS_z + U\sum_{j=-\infty}^{\infty} n_{j\uparrow}n_{j\downarrow}. \tag{1}$$

The fermionic creation, $\psi_{j\alpha}^\dagger$, and annihilation, $\psi_{j\alpha}$, operators ($\alpha$ is a spin index, $\alpha = \uparrow,\downarrow$) satisfy canonical equal-time anti-commutation relations,

$$\psi_{j\alpha}\psi_{j'\alpha'}^\dagger + \psi_{j'\alpha'}^\dagger \psi_{j\alpha} = \delta_{jj'}\delta_{\alpha\alpha'}, \tag{2}$$

where

$$\delta_{ab} = \begin{cases} 1, & a = b, \\ 0, & a \neq b, \end{cases} \tag{3}$$

is the Kronecker delta symbol. The operator $n_{j\alpha} = \psi_{j\alpha}^\dagger \psi_{j\alpha}$ is the density operator for the spin-up ($\alpha = \uparrow$) and spin-down ($\alpha = \downarrow$) fermions, and

$$n_j = n_{j\uparrow} + n_{j\downarrow} \tag{4}$$

counts the total density of fermions on site $j$. The local spin vector $\mathbf{s}(j) = (s_x(j), s_y(j), s_z(j))$ is defined as

$$\mathbf{s}(j) = \frac{1}{2} \begin{pmatrix} \psi_{j\uparrow}^\dagger & \psi_{j\downarrow}^\dagger \end{pmatrix} \boldsymbol{\sigma} \begin{pmatrix} \psi_{j\uparrow} \\ \psi_{j\downarrow} \end{pmatrix}, \tag{5}$$

where

$$\boldsymbol{\sigma} = (\sigma_x, \sigma_y, \sigma_z) \tag{6}$$

is the vector composed of the three Pauli matrices. In particular, $s_z(j) = (n_{j\uparrow} - n_{j\downarrow})/2$. The spin-ladder operators $s_\pm(j) = s_x(j) \pm i s_y(j)$ flip the $z$ component of a local spin, and read $s_+(j) = \psi_{j\uparrow}^\dagger \psi_{j\downarrow}$ and $s_-(j) = \psi_{j\downarrow}^\dagger \psi_{j\uparrow}$, respectively. The total number of particles,

$$N = \sum_{j=-\infty}^{\infty} n_j, \tag{7}$$

and the $z$ projection of the total spin,

$$S_z = \sum_{j=-\infty}^{\infty} s_z(j), \tag{8}$$

are conserved quantities.

In the present work, we focus on the infinitely strong repulsion limit, $U \to \infty$, of the Hubbard model (1). It would cost infinite energy to put two particles on any site in this limit, due to the on-site interaction term $U\sum_j n_{j\uparrow}n_{j\downarrow}$ in Eq. (1). We thus arrive at the no double occupancy (NDO) constraint, which can be fulfilled by applying the projection operator

$$P = \prod_{j=-\infty}^{\infty} (1 - n_{j\uparrow}n_{j\downarrow}) \tag{9}$$

to the Hamiltonian (1). This results in the $t-0$ model [71],

$$H = P \left[ -\sum_{\substack{j=-\infty \\ \alpha=\uparrow,\downarrow}}^{\infty} (\psi_{j\alpha}^\dagger \psi_{j+1\alpha} + \psi_{j+1\alpha}^\dagger \psi_{j\alpha}) - hN + 2BS_z \right] P. \tag{10}$$

Each site of the lattice can now be either empty, or occupied by one spin-up or one spin-down fermion.

Any eigenstate of the Hamiltonian (1) can be constructed of basis states

$$|\mathbf{j}, \boldsymbol{\alpha}\rangle = \psi^{\dagger}_{j_1, \alpha_1} \dots \psi^{\dagger}_{j_N, \alpha_N} |0\rangle, \qquad j_1 \leq j_2 \dots \leq j_N, \tag{11}$$

where $|0\rangle$ is the vacuum state, which contains no fermions. Only those states satisfying

$$P|\mathbf{j}, \boldsymbol{\alpha}\rangle \neq 0, \tag{12}$$

which is equivalent to the NDO constraint

$$j_1 < j_2 \dots < j_N, \tag{13}$$

can be used to construct the eigenstates of the Hamiltonian (10). Taking the coordinates $j_1, \dots, j_N$ and the spin orientations $\alpha_1, \dots, \alpha_N$ from a state (11) satisfying the NDO constraint we define the state

$$|f\rangle = c^{\dagger}_{j_1} \dots c^{\dagger}_{j_N} |0\rangle \tag{14}$$

made of spinless fermions ($c^{\dagger}_j$ creates, and $c_j$ annihilates a fermion on site $j$), and the state

$$|\ell\rangle = |\alpha_1, \dots, \alpha_N\rangle \tag{15}$$

of a spin-1/2 chain of length $N$ uniquely. The reverse is also true: having defined $|f\rangle \neq 0$ by Eq. (14) and $|\ell\rangle$ by Eq. (15) one can reconstruct $|\mathbf{j}, \boldsymbol{\alpha}\rangle$, which will satisfy the NDO constraint. Thus, we can write

$$|\mathbf{j}, \boldsymbol{\alpha}\rangle = |f\rangle \otimes_f |\ell\rangle, \qquad j_1 < j_2 < \dots < j_N. \tag{16}$$

The subscript $f$ in $\otimes_f$ indicates that the tensor product $\otimes$ is equipped with a constraint: the number of spinless fermions in the charge part of the wave function, $|f\rangle$, determines the number of sites of the spin chain in the spin part of the wave function, $|\ell\rangle$.

The operators $\psi^{\dagger}_{j\alpha}$ and $\psi_{j\alpha}$ can be expressed via $c^{\dagger}_j$, $c_j$, and the local spin operators

$$\ell(m) = 1 \otimes \dots \otimes \frac{\sigma(m)}{2} \otimes \dots \otimes 1 \tag{17}$$

acting onto $|\ell\rangle$, where $\boldsymbol{\sigma}$ is defined by Eq. (6). The explicit formulas are given in Ref. [52]. The local spin operators (5) preserve the fermion number $N$, and their representation is consequently simpler [72]. Its key ingredient is the counting operator

$$\mathcal{N}_j = \sum_{a=-\infty}^{j} n_a. \tag{18}$$

The value of $\mathcal{N}_j$ increases by one each time a lattice site is occupied, when $j$ runs from minus infinity to infinity. The local density operator (4) expressed via spinless fermion operators read

$$n_j = c^{\dagger}_j c_j. \tag{19}$$

The local spin operator can be represented via $\ell(m)$ and $n_j$ using Eq. (18):

$$\mathbf{s}(j) = n_j \sum_{m=-\infty}^{\infty} \ell(m) \delta_{m, \mathcal{N}_j}. \tag{20}$$

Let us illustrate how Eq. (20) works for $|\Psi\rangle = \psi^{\dagger}_{1, \alpha_1} \psi^{\dagger}_{5, \alpha_2} |0\rangle$. Following Eq. (16) we write $|\Psi\rangle = c^{\dagger}_1 c^{\dagger}_5 |0\rangle \otimes_f |\alpha_1, \alpha_2\rangle$. Applying $\mathbf{s}(j)$ to $|\Psi\rangle$ we get zero for $j$ other than one and five,

because of vanishing $n_j$ (naturally, there are no spins at the lattice sites not occupied by the fermions). We have $n_j = 1$ for sites one and five; $\mathcal{N}_1 = 1$ and $\mathcal{N}_5 = 2$ imply $\mathbf{s}(1) = \boldsymbol{\ell}(1)$ and $\mathbf{s}(5) = \boldsymbol{\ell}(2)$, respectively, and the action of the operator $\boldsymbol{\ell}$ is defined by Eq. (17).

We rewrite Eq. (20) as

$$\mathbf{s}(j) = \sum_{m=-\infty}^{\infty} \int_{-\pi}^{\pi} \frac{d\lambda}{2\pi} \, n_j e^{i\lambda(\mathcal{N}_j - m)} \boldsymbol{\ell}(m) \,, \tag{21}$$

using the integral representation of the Kronecker delta symbol. The Hamiltonian (10) expressed via $c_j^\dagger$, $c_j$, and $\boldsymbol{\ell}(m)$ reads

$$H = -\sum_{j=-\infty}^{\infty} (c_j^\dagger c_{j+1} + c_{j+1}^\dagger c_j) - hN + 2BS_z \,, \tag{22}$$

where $N$ is the number operator (7) written via the spinless fermion density (19). We have

$$S_z|\Psi\rangle = |f\rangle \otimes_f L_z|\ell\rangle \,, \qquad L_z = \sum_{m=1}^{N} \ell_z(m) \,. \tag{23}$$

The eigenbasis of the Hamiltonian (22) is formed by the vectors $|\mathbf{k}\rangle \otimes_f |\ell\rangle$, where

$$|\mathbf{k}\rangle = c_{k_1}^\dagger \dots c_{k_N}^\dagger |0\rangle \tag{24}$$

are the momentum-space components of the vector (14), and

$$c_j^\dagger = \frac{1}{\sqrt{2\pi}} \sum_k e^{-ikj} c_k^\dagger \,. \tag{25}$$

Therefore,

$$H\left(|\mathbf{k}\rangle \otimes_f |\ell\rangle\right) = (E + E_\ell)\left(|\mathbf{k}\rangle \otimes_f |\ell\rangle\right) \,, \tag{26}$$

where

$$E = \sum_{i=1}^{N} \varepsilon(k_i) \,, \qquad \varepsilon(k) = -2\cos(k) \,, \tag{27}$$

and

$$E_\ell = -h(N_\uparrow + N_\downarrow) + B(N_\uparrow - N_\downarrow) \,. \tag{28}$$

Here, $|\ell\rangle$ is a state of a spin chain containing $N_\uparrow$ spin-up and $N_\downarrow = N - N_\uparrow$ spin-down sites.

The use of the representation (16) for the $t-0$ model is called the spin-charge separation in some literature [73]. A few words of caution should be mentioned about this terminology. Indeed, $|f\rangle$ and $|\ell\rangle$ can be chosen independently from each other, with the only constraint defining the length of the spin chain via the fermion number $N$. A separation can also be seen in the Hamiltonian (22): $S_z$ acts non-trivially onto $|\ell\rangle$, Eq. (23), the remaining terms act onto $|f\rangle$, and $L_z$ depends on $N$. However, Eq. (21) and the formulas for $\psi_{j\alpha}^\dagger$ and $\psi_{j\alpha}$, Ref. [52], cannot be split into a product of operators containing only spin, $\boldsymbol{\ell}(m)$, and charge, $c_j^\dagger$ and $c_j$, parts. Although the bosonization offers splitting of the local operators into spin and charge parts at low energies and momenta (this procedure is also called the spin-charge separation in the literature), it requires the linearity of the excitation spectrum [74,75]. Thus, the spin-charge separation understood in the sense of the transformation (16)–(22) works far beyond the bosonization in the model (10). It captures, in particular, the polaron [72,76] and the spin-incoherent [77] physics of the model. A limitation of the transformation (16)–(22) is the need for the NDO constraint, resulting in its failure for the finite $U$ Hubbard model, Eq. (1), where the bosonization works. There exists a transformation aimed to separate spin and charge degrees of freedom for the finite $U$ Hubbard model beyond the bosonization paradigm [60, 61,78–82], but its analysis lies out of the scope of the present work.

# 3 Dynamical correlation functions

In this section we evaluate the connected, two point dynamical correlation function of the $z$-projection of spins,

$$\sigma^{(c)}(j-j',t) = \langle s_z(j,t)s_z(j',0)\rangle_T - \langle s_z(j,t)\rangle_T \langle s_z(j',0)\rangle_T. \tag{29}$$

The average

$$\langle\cdots\rangle_T = \frac{1}{Z}\sum_{N=0}^{\infty}\sum_{f,\ell}\Big(\langle\ell|\otimes_f\langle f|\Big)e^{-\beta H}\cdots\Big(|f\rangle\otimes_f|\ell\rangle\Big) \tag{30}$$

is computed in the grand canonical ensemble at temperature $T$, chemical potential $h$, and magnetic field $B$. Note that the right hand side of this expression is the trace of the equilibrium density matrix $e^{-\beta H}/Z$, where $Z$ is the grand partition function, and $\beta = 1/T$ is the inverse temperature. The trace is invariant with respect to the choice of the basis, therefore $\sum_f\langle f|\cdots|f\rangle$ can be replaced with $\sum_{\mathbf{k}}\langle\mathbf{k}|\cdots|\mathbf{k}\rangle$, where $|\mathbf{k}\rangle$ is defined by Eq. (24). We represent the function (29) as a Fredholm determinant of an integrable integral operator. This representation is exact for any value of relative coordinate $j-j'$ and time $t$.

## 3.1 Local magnetization

We start evaluating Eq. (29) with considering the local magnetization $\langle s_z(j,t)\rangle_T$, which does not depend on time at equilibrium. We substitute the representation (21) into Eq. (30) and calculate $\sum_{\ell}\langle\ell|\cdots|\ell\rangle$ in the first place:

$$\sum_{\ell}e^{-\beta E_{\ell}} = e^{\beta hN}[2\cosh(\beta B)]^N, \tag{31}$$

and

$$\sum_{\ell}e^{-\beta E_{\ell}}\langle\ell|\ell_z(m)|\ell\rangle = -\frac{1}{2}\tanh(\beta B)e^{\beta hN}[2\cosh(\beta B)]^N. \tag{32}$$

We see that the right hand side of Eq. (32) is independent of $m$. Taking into account that

$$\sum_{m=-\infty}^{\infty}e^{i\lambda m} = 2\pi\delta(\lambda), \qquad -\pi\le\lambda\le\pi, \tag{33}$$

we arrive at the sum over spinless fermion states, which we write in the basis (24):

$$\langle s_z(j,t)\rangle_T = -\frac{\tanh(\beta B)}{2}\frac{\sum_{N=0}^{\infty}\sum_{\mathbf{k}}e^{-\beta\tilde{E}_{\mathbf{k}}}\langle\mathbf{k}|n_j|\mathbf{k}\rangle}{\sum_{N=0}^{\infty}\sum_{\mathbf{k}}e^{-\beta\tilde{E}_{\mathbf{k}}}}. \tag{34}$$

The energy $\tilde{E}_{\mathbf{k}}$ is a sum of single-particle energies $\tilde{\varepsilon}(k_i)$ which are shifted relative to $\varepsilon(k_i)$ defined by Eq. (27):

$$\tilde{E}_{\mathbf{k}} = \sum_{i=1}^{N}\tilde{\varepsilon}(k_i), \qquad \tilde{\varepsilon}(k) = \varepsilon(k) - h - \frac{\log[2\cosh(\beta B)]}{\beta}. \tag{35}$$

Indeed, such a modification accounts for the charge-dependent prefactors in the spin average (32). After accounting for these subtleties the rest of the computations are performed as in a free Fermi gas and result in the following expression in the thermodynamic limit

$$\langle s_z(j,t)\rangle_T = -\frac{\tanh(\beta B)}{2}\int_{-\pi}^{\pi}\frac{dk}{2\pi}n_{\rho}(k), \tag{36}$$

where $n_\rho(k)$ is a Fermi-Dirac distribution with the modified energies

$$n_\rho(k) = \frac{1}{e^{\beta \tilde{\varepsilon}(k)} + 1} = \frac{2\cosh(\beta B)}{2\cosh(\beta B) + e^{\beta[\varepsilon(k)-h]}}. \tag{37}$$

## 3.2 The two-point function

Now we turn to the two-point correlation function

$$\sigma(j - j', t) = \langle s_z(j, t) s_z(j', 0) \rangle_T. \tag{38}$$

Using the same arguments and employing presentation (21) we factorize the average in Eq. (38) into the spin and charge sectors

$$\sigma(j-j', t) = \frac{1}{Z} \sum_{N=0}^{\infty} \sum_{\mathbf{k}, \ell} \sum_{m,m'=-\infty}^{\infty} \int_{-\pi}^{\pi} \frac{d\lambda}{2\pi} \frac{d\lambda'}{2\pi} e^{-i\lambda m + i\lambda' m'} e^{-\beta E_{\mathbf{k}}} \mathcal{C}_p(\lambda, \lambda'; j-j'; t) \mathcal{S}(m, m'). \tag{39}$$

Similar separation formulas, though approximate, appear in the desription of the tracer dynamics [83]. The charge part is the correlation function of the free spinless fermions

$$\mathcal{C}_p(\lambda, \lambda'; j-j'; t) = \langle \mathbf{k} | n_j(t) e^{i\lambda \mathcal{N}_j(t)} e^{-i\lambda' \mathcal{N}_{j'}(0)} n_{j'}(0) | \mathbf{k} \rangle. \tag{40}$$

The spin part formally is defined as

$$\mathcal{S}(m, m') = e^{-\beta E_\ell} \langle \ell | s_z(m) s_z(m') | \ell \rangle. \tag{41}$$

Notice that here the time dependence canceled out since $s_z(m)$ does commute with the Hamiltonian. For the chains of length $N$, similarly to (32) we can write

$$\sum_{\ell} \mathcal{S}(m, m') = \text{Tr}\left[ s_z(m) s_z(m') e^{-\beta(2S_z B - hN)} \right] = \frac{1}{4} e^{\beta hN} (2\cosh\beta B)^N \left( \frac{\delta_{m,m'}}{\cosh^2 \beta B} + \tanh^2(\beta B) \right). \tag{42}$$

Using relation (33) we arrive at the following representation for the total correlation function

$$\sigma(j - j', t) = \sigma_0(j - j', t) + \sigma_1(j - j', t), \tag{43}$$

where

$$\sigma_0(j - j', t) = \frac{\tanh^2(\beta B)}{4Z} \sum_{N=0}^{\infty} \sum_{\mathbf{k}} e^{-\beta \tilde{E}_{\mathbf{k}}} \langle \mathbf{k} | n_j(t) n_{j'}(0) | \mathbf{k} \rangle, \tag{44}$$

$$\sigma_1(j - j', t) = \frac{1}{4\cosh^2(\beta B)} \frac{1}{Z} \sum_{N=0}^{\infty} \sum_{\mathbf{k}} e^{-\beta \tilde{E}_{\mathbf{k}}} \int_{-\pi}^{\pi} \frac{d\lambda}{2\pi} \langle \mathbf{k} | n_j(t) e^{i\lambda \mathcal{N}_j(t)} e^{-i\lambda \mathcal{N}_{j'}(0)} n_{j'}(0) | \mathbf{k} \rangle. \tag{45}$$

Here as above the energy $\tilde{E}_{\mathbf{k}}$ is constructed from the quasienergies (35).

The first contribution $\sigma_0$ can be computed immediately by applying the Wick's theorem, but instead we proceed with the computation of $\sigma_1$ and then take limit $\lambda \to 0$ of the function under the integral. To compute the average in $\sigma_1$ we notice that $e^{i\lambda n_j} n_j = e^{i\lambda} n_j$ and

$$n_j(t) = \frac{e^{i\lambda n_j(t)} - 1}{e^{i\lambda} - 1}, \qquad n_{j'}(0) = \frac{e^{-i\lambda n_{j'}(0)} - 1}{e^{-i\lambda} - 1}. \tag{46}$$

This way if we have a string correlator

$$\mathcal{F}_\lambda^{(\mathbf{k})}(j - j', t) = \langle \mathbf{k} | e^{i\lambda \mathcal{N}_j(t)} e^{-i\lambda \mathcal{N}_{j'}(0)} | \mathbf{k} \rangle, \tag{47}$$

then

$$\langle\mathbf{k}|n_j(t)e^{i\lambda\mathcal{N}_j(t)}e^{-i\lambda\mathcal{N}_{j'}(0)}n_{j'}(0)|\mathbf{k}\rangle = \frac{2\mathcal{F}_\lambda^{(\mathbf{k})}(j-j',t)-\mathcal{F}_\lambda^{(\mathbf{k})}(j-j'-1,t)-\mathcal{F}_\lambda^{(\mathbf{k})}(j-j'+1,t)}{2(1-\cos\lambda)}. \quad (48)$$

The string correlator $\mathcal{F}_\lambda^{(\mathbf{k})}(j-j',t)$ can be expressed as a single determinant, which in the thermodynamic limit takes a form of a Fredholm determinant.

$$\mathcal{F}_\lambda(x,t) = \det(1+\hat{\mathcal{U}}). \quad (49)$$

The kernel of the operator $\hat{\mathcal{U}}$ reads

$$\mathcal{U}(k,q) = \frac{\ell_+(x,k)\ell_-(x,q)-\ell_-(x,k)\ell_+(x,q)}{2\pi\sin\frac{k-q}{2}}, \quad (50)$$

where

$$\ell_+(x,k) = \sqrt{n_\rho(k)}\left(\frac{1-\cos\lambda}{2}E_+(x,k)+\frac{\sin\lambda}{2}E_-^{-1}(x,k)\right), \quad (51)$$

$$\ell_-(x,k) = E_-(x,k)\sqrt{n_\rho(k)}, \qquad E_-(k) = e^{it\varepsilon(k)/2-ixk/2}, \quad (52)$$

with

$$E_+(x,k) = E(x,k)E_-(x,k), \quad (53)$$

$$E(x,k) = \int_{-\pi}^{\pi}\frac{dq}{2\pi}\frac{e^{-it\varepsilon(q)+ixq}}{\tan\frac{q-k}{2}}. \quad (54)$$

There integral is taken in the principal value sense. We present the derivation in Appendix (A) (see also [84]).

Taking into account a special structure of the coordinate dependence in (48), it is useful to introduce the shift operator $\hat{S}$ acting on the functions of the discrete variable $x$

$$\hat{S}f(x) = 2f(x)-f(x+1)-f(x-1), \quad (55)$$

which is nothing but a discrete analog of the second derivative. This way, $\sigma_1$ in the thermodynamic limit reads

$$\sigma_1(x,t) = \frac{1}{4\cosh^2(\beta B)}\int_{-\pi}^{\pi}\frac{d\lambda}{2\pi}\frac{\hat{S}\mathcal{F}_\lambda(x,t)}{2(1-\cos\lambda)}, \quad (56)$$

and $\sigma_0$ can be presented as

$$\sigma_0(x,t) = \frac{\tanh^2(\beta B)}{4}\frac{\hat{S}\mathcal{F}_\lambda(x,t)}{2(1-\cos\lambda)}\Big|_{\lambda=0}. \quad (57)$$

Expanding Fredholm determinants (see Appendix (B)) we obtain the following expressions

$$\sigma_0(x,t)$$
$$= \frac{\tanh^2(\beta B)}{4}\left(\left[\int_{-\pi}^{\pi}\frac{dk}{2\pi}n_\rho(k)\right]^2+\int_{-\pi}^{\pi}\frac{dk}{2\pi}n_\rho(k)e^{it\varepsilon(k)-ikx}\int_{-\pi}^{\pi}\frac{dq}{2\pi}e^{-it\varepsilon(q)+iqx}(1-n_\rho(q))\right). \quad (58)$$

This way, taking into account (36) the connected correlation function (29) reads

$$\sigma^{(c)}(x,t) = \sigma_0^{(c)}(x,t)+\sigma_1(x,t), \quad (59)$$

with

$$\sigma_0^{(c)}(x,t) = \frac{\tanh^2(\beta B)}{4} \int\limits_{-\pi}^{\pi} \frac{dk}{2\pi} n_\rho(k) e^{it\varepsilon(k)-ikx} \int\limits_{-\pi}^{\pi} \frac{dq}{2\pi} e^{-it\varepsilon(q)+iqx}(1-n_\rho(q)). \tag{60}$$

The formula (59) is an exact form for the spin-spin correlation function (29) in the thermo-dynamic limit. Fredholm determinants can be effectively evaluated numerically [85] at any values of $x$ and $t$. The universal physical characteristics can be extracted from (59) by studying its asymptotic behavior for large $x$ and $t$. We perform this analysis in the next chapter.

## 4 Transport coefficients

Having derived the two-point function now we turn to its asymptotic analysis. This way we derive the key transport properties of the model. We show that in the general case the model supports both ballistic and diffusive spin transport, and we derive the characteristic quantities, the Drude weight and the diffusion constant. We use the notations of Ref. [86].

Let us start with the static covariance defined as

$$\mathsf{C} = \sum_x \sigma^{(c)}(x, t=0). \tag{61}$$

First, we simplify the kernel of the Fredholm determinant. Integral in (54) can be evaluated exactly

$$E(x,k)\Big|_{t=0} = i\,\mathrm{sgn}(x)e^{ikx}, \tag{62}$$

so the full kernel (50) simplifies into

$$\mathcal{U}(k,q) = \frac{e^{i\lambda\,\mathrm{sgn}(x)}-1}{2\pi} \sqrt{n_\rho(k)} \frac{\sin\frac{|x|(k-q)}{2}}{\sin\frac{k-q}{2}} \sqrt{n_\rho(q)}. \tag{63}$$

In this form, this kernel is identical to one of the effective fermions with the constant phase shift $\lambda$ [68]. The Fredholm determinant $\mathcal{F}_\lambda$ can be considered as a series expansion of the traces of the antisymmetric powers of the $\hat{\mathcal{U}}$. The first few terms read

$$\mathcal{F}_\lambda = 1 + (e^{i\lambda\,\mathrm{sgn}(x)}-1)|x| \int\limits_{-\pi}^{\pi} \frac{dk}{2\pi} n_\rho(k) + O((e^{i\lambda\,\mathrm{sgn}(x)}-1)^2). \tag{64}$$

Taking into account that for $n > 1$

$$\frac{(e^{\pm i\lambda}-1)^n}{1-\cos\lambda} = -2e^{\pm i\lambda}(e^{\pm i\lambda}-1)^{n-2}, \tag{65}$$

we see that terms in the remainder in (64) vanish after the integration over $\lambda$. Further, we compute the action of the shift operator on the first two terms.

$$\hat{S}1 = 0, \qquad \hat{S}(e^{i\lambda\,\mathrm{sgn}(x)}-1)|x| = 2(1-\cos\lambda)\delta_{x,0}. \tag{66}$$

Therefore performing summation over $x$ we obtain contribution to the static covariance from $\sigma_1$

$$\sum_x \sigma_1(x,t=0) = \frac{1}{4\cosh^2(\beta B)} \int\limits_{-\pi}^{\pi} \frac{dk}{2\pi} n_\rho(k). \tag{67}$$

Now let us turn to an evaluation of $\sigma_0^c$ part in (59). Using (33) we arrive at

$$C_b \equiv \sum_x \sigma_0^c(x, t=0) = \frac{\tanh^2(\beta B)}{4} \int_{-\pi}^{\pi} \frac{dk}{2\pi} n_\rho(k)(1-n_\rho(k)). \tag{68}$$

Notice that this evaluation remains valid even at $t \neq 0$. The same statement can be demonstrated even for $\sigma_1$, using the fact that $\hat{S}\mathcal{F}$ plays a role of second derivative, so after the summation over $x$, one has to take into account only boundary terms at large distances for which one can use the asymptotic in the space like regime (see for instance [67–69]). Overall, we obtain

$$C = \sum_x \sigma^{(c)}(x, t) = \frac{1}{4} \int_{-\pi}^{\pi} \frac{dk}{2\pi} n_\rho(k) - \frac{\tanh^2(\beta B)}{4} \int_{-\pi}^{\pi} \frac{dk}{2\pi} n_\rho(k)^2. \tag{69}$$

Further, following [86], we define the Drude weight D and Onsager matrix $\mathfrak{L}$ via the asymptotic at long times of the second moment, namely

$$\frac{1}{2} \sum_x x^2 \left( \sigma^{(c)}(x, t) + \sigma^{(c)}(x, -t) \right) = Dt^2 + \mathfrak{L}t + o(t). \tag{70}$$

To account for the contributions from $\sigma_0$ we use second derivative of the relation (33) to arrive at

$$\frac{1}{2} \sum_x x^2 \left( \sigma_0^{(c)}(x, t) + \sigma_0^{(c)}(x, -t) \right) = Dt^2 + o(t), \tag{71}$$

with

$$D = \frac{\tanh^2(\beta B)}{4} \int_{-\pi}^{\pi} \frac{dk}{2\pi} \varepsilon'(k)^2 n_\rho(k)(1-n_\rho(k)). \tag{72}$$

More specifically, we can describe not only the second moment but the full asymptotic behavior of $\sigma_0(x, t)$ on the ballistic scale $x, t \to \infty$ and $x/t = $ const. For $x > 2t$ the integrals vanish exponentially, so

$$\sigma_0(x, t) = O(e^{-\#x}), \tag{73}$$

while for $0 < x < 2t$ they are dominated by two saddle points $k_0 = \arcsin(x/2t)$ and $k_1 = \pi - k_0$. This way introducing

$$n_\pm = \frac{2\cosh(\beta B)}{2\cosh(\beta B) + e^{-\beta(h \pm 2\sqrt{1-x^2/(2t)^2})}}, \tag{74}$$

we obtain

$$\sigma_0(x, t) \approx \frac{\tanh^2(\beta B)}{4} \sum_{s=\pm} \frac{n_s(1-n_s+(-1)^x(1-n_{-s})e^{-2is\Phi})}{\sqrt{(2\pi)((2t)^2-x^2)}}, \tag{75}$$

with

$$\Phi = \sqrt{(2t)^2 - x^2} + xk_0 - \frac{i\pi}{4}. \tag{76}$$

The integral of the Fredholm determinants in $\sigma_1$ are expected to produce diffusive terms in the region $x \sim \sqrt{t}$. To proceed with the asymptotic of the determinant we notice that the kernel (50) also appears in the correlation function of one-dimensional impenetrable anyons upon the identification $\gamma\theta(k) = n_F(k) = n_\rho(k)$ [67,87,88]. Moreover, this kernel is nothing but a generalized sine-kernel on a lattice so its asymptotic behavior can be found rigorously

by solving the corresponding Riemann-Hilbert problem [87, 89], or obtained heuristically, by using the effective form factors approach [67–69]. The result for $x < 2t$ reads

$$\mathcal{F}_\lambda(x,t) \approx \frac{C(x/t)}{t^{(\delta v)^2}} \exp\left( \int_{-\pi}^{\pi} \frac{dq}{2\pi} |x - \varepsilon'(q)t| \log(1 + n_\rho(q)(e^{i\lambda \operatorname{sgn}(x - \varepsilon'(q)t)} - 1)) \right). \tag{77}$$

Here

$$\delta v \sim \frac{\log(1 + n_\rho(k_*)(e^{i\lambda} - 1))}{2\pi i} - \left( -\frac{\log(1 + n_\rho(k_*)(e^{-i\lambda} - 1))}{2\pi i} \right), \tag{78}$$

with $k_*$ is one of the critical points $k_0$ or $\pi - k_0$ introduced after Eq. (73). In principle, we have to sum over all these points, however further we will see that the integral is dominated by $\lambda \sim x/t \sim 1/\sqrt{t}$, therefore the power-law prefactors are of the order $t^{-(\delta v)^2} \sim \exp\left( O((\log t)/\sqrt{t}) \right)$, so we can regard them to be constant as well as the prefactor $C(x/t) \approx C(0)$. We are going to compute integral in (56) by means of the saddle point methods. For this let us expand expression in the exponential for small $\lambda$

$$\log \mathcal{F}_\lambda(x,t) \approx i\lambda x \int \frac{dq}{2\pi} n_\rho(q) - \frac{\lambda^2}{2} t \int \frac{dq}{2\pi} |\varepsilon'(q)| n_\rho(q)(1 - n_\rho(q)). \tag{79}$$

Here we assume that $x \sim \sqrt{t}$ or less. We also assume that due to the symmetric properties of $\varepsilon(q)$ and $n_\rho(q)$ (see (37)), we have

$$\int \varepsilon'(q) n_\rho(q) dq = 0. \tag{80}$$

So after integration over $\lambda$ we obtain

$$\sigma_1(x,t) = \frac{C(0) \int_{-\pi}^{\pi} \frac{dq}{2\pi} n_\rho(q)}{4 \cosh^2(\beta B)} \frac{e^{-x^2/(2\mathcal{D}t)}}{\sqrt{2\pi \mathcal{D}t}}, \tag{81}$$

with

$$\mathcal{D} = \frac{\int_{-\pi}^{\pi} |\varepsilon'(q)| n_\rho(q)(1 - n_\rho(q)) \frac{dq}{2\pi}}{\left[ \int_{-\pi}^{\pi} n_\rho(q) \frac{dq}{2\pi} \right]^2}. \tag{82}$$

In Fig. 1 we compare theoretical predictions (81) with numerical results obtained from the exact expression (56) using numerical methods described in [85]. This allows us to compute non only the diffusion constant $\mathcal{D}$ but also conclude that the constant $C(0) \approx 1$ for various regimes.

Fitting the numerically evaluated (56) by the function $\exp(-x^2/(2\mathcal{D}t) + B)$ at the space scale $x \sim \sqrt{t}$ we could estimate the diffusion constant at the finite time scales. The results are shown in the inset in Fig. 1. We see that the "infinite time" limit is reached very quickly.

For $B = 0$, or in the case of infinite temperature, the spin-spin correlation function is given only by $\sigma_1$ and has a diffusive shape. Then the condition $C(0) = 1$ comes naturally via the connection with the initial profile. For infinite temperature $n_\rho(q) = \rho$ and

$$\mathcal{D} = 2(\rho^{-1} - 1)/\pi. \tag{83}$$

In the absence of magnetic fields we have $\rho = 2/3$, thus

$$\mathcal{D} = 1/\pi. \tag{84}$$

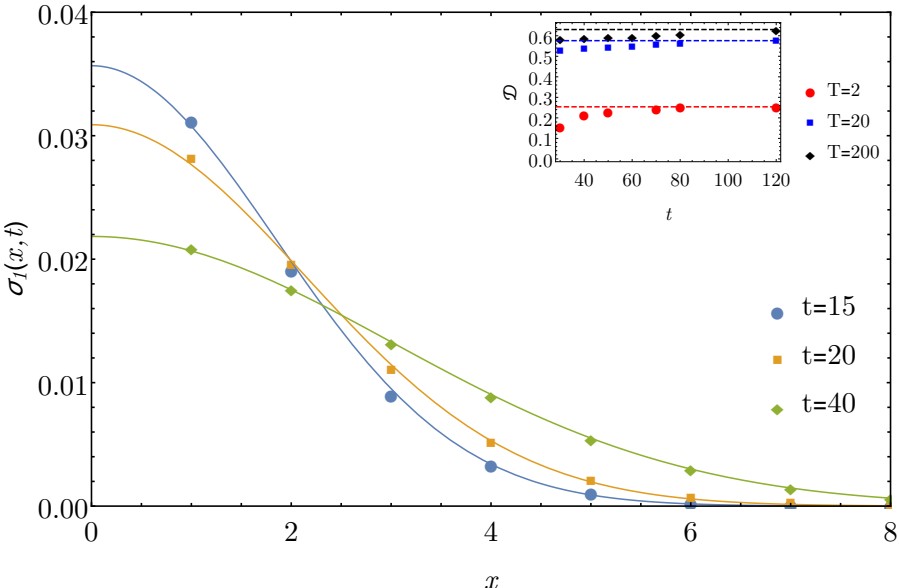

Figure 1: Coordinate dependence of the diffusive part of the spin-spin correlation function. Solid lines show analytic answer (81) and dots correspond to numeric evaluation of (56) for the density given by (37) with $h = 2$, $B = 1$, $T = 2$, for times shown in legends. Inset shows the diffusion constant $\mathcal{D}$ after fitting results of (56), for $B = 0$, $h = 2$ and temperatures according to the legend. Dashed lines show analytic answer (82).

This coincides with the results obtained with the tracer dynamics in [83]. Note that the normalization of the Hamiltonian in [83] includes an extra factor of $1/2$ (see eq. (48) in that work), therefore their diffusion constant differs from ours also in a factor of $1/2$.

The magnetic field dependence for various temperatures is depicted in Fig. (2).

Notice that if we formally replace summation into integration with the profile (81) and put $C(0) = 1$ we recover the static correlation result (67). Similarly, we can compute the Onsager matrix $\mathfrak{L}$ in (70)

$$\mathfrak{L} = \frac{\int\limits_{-\pi}^{\pi} |\varepsilon'(q)| n_\rho(q)(1 - n_\rho(q)) \frac{dq}{2\pi}}{4\cosh^2(\beta B) \int\limits_{-\pi}^{\pi} n_\rho(q) \frac{dq}{2\pi}} . \tag{85}$$

We observe that the diffusion constant $\mathfrak{D} = \mathfrak{L}/C$ coincides with $\mathcal{D}$ only when the ballistic part is absent (i.e. for $B = 0$).

## 5 Thermodynamics of the model and GHD diffusion constant

In our approach, we did not have to introduce the Euler and the diffusive scales, but they appear naturally from the exact expressions. Nevertheless, we can also compare our results with predictions of the generalized hydrodynamics. The essential ingredient to it is Thermodynamic Bethe Ansatz (TBA) formulation, which we briefly recall below.

Let us start with Bethe Ansatz equations for the Hamiltonian (10). In notations of [59] every eigenstate is parameterized by $N$ unequal quasimomenta $k_1, \ldots k_N$, and by the set of $M$

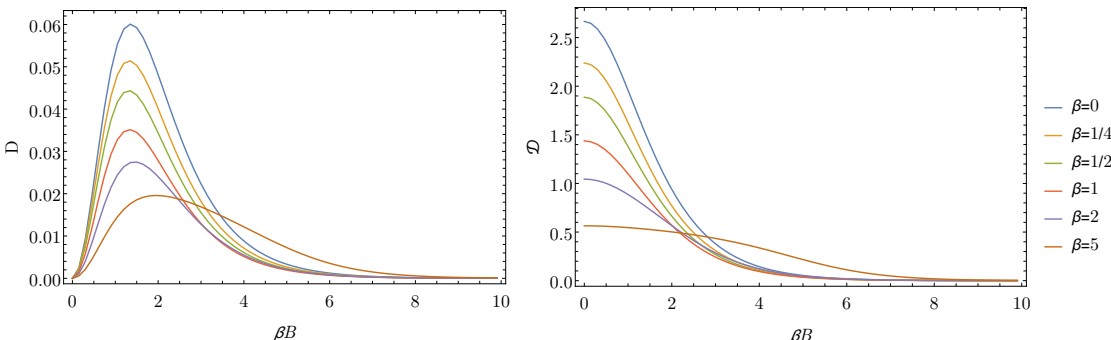

Figure 2: Magnetic field dependence of the Drude weight and the diffusion constant for various temperatures, $h = 1$.

auxiliary momenta $\lambda_1 \dots \lambda_M$, satisfying

$$e^{ik_a L} = e^{i\Lambda}, \quad a = 1, \dots N, \tag{86}$$

$$e^{i\lambda_b N} = (-1)^{M+1}, \quad b = 1, \dots M, \qquad \Lambda = \sum_{b=1}^{M} \lambda_b. \tag{87}$$

The corresponding state has $M$ spins down and $N - M$ spins up. To formulate these equations as TBA, we introduce the corresponding densities of the quasiparticles

$$\rho_p(k_i) = \frac{1}{L(k_{i+1} - k_i)}, \qquad \sigma_p(\lambda_j) = \frac{1}{L(\lambda_{j+1} - \lambda_j)}. \tag{88}$$

Then the corresponding energy density reads

$$\frac{E}{L} = \int_{-\pi}^{\pi} (\varepsilon(k) - h)\rho_p(k)dk + B\left(\int_{-\pi}^{\pi} \rho_p(k)dk - 2\int_{-\pi}^{\pi} \sigma_p(\lambda)d\lambda\right). \tag{89}$$

The total densities are constant

$$\rho_t(k) = \frac{1}{2\pi}, \qquad \sigma_t(\lambda) = \int_{-\pi}^{\pi} \rho_p(k)\frac{dk}{2\pi}. \tag{90}$$

Notice that the only term describing an interaction between quasimomenta and auxiliary momenta comes in the normalization for the latter. In other aspects both these particles can be considered as fermions, so the free energy takes the following form

$$F = LE - TLs(\rho_t, \rho_p) - TLs(\sigma_t, \sigma_p), \tag{91}$$

with

$$s(\rho_t, \rho_p) = \int_{-\pi}^{\pi} dk(\rho_t \log \rho_t - \rho_p \log \rho_p - \bar{\rho}_p \log \bar{\rho}_p), \qquad \bar{\rho}_p = \rho_t - \rho_p, \tag{92}$$

and identically for $s(\sigma_t, \sigma_p)$. To describe thermodynamic equilibrium we compute variations over $\rho_p$ and $\sigma_p$, which leads to the following equations, correspondingly

$$\varepsilon(k) - h - B - T\log\frac{\rho_p}{\rho_t - \rho_p}(k) - \frac{T}{2\pi}\int_{-\pi}^{\pi}\log\frac{\sigma_t}{\sigma_t - \sigma_p(\lambda)}d\lambda = 0, \tag{93}$$

$$2B + T \log \frac{\sigma_p(\lambda)}{\sigma_t - \sigma_p(\lambda)} = 0 \,, \tag{94}$$

which leads to

$$n_\sigma(\lambda) = \frac{\sigma_p(\lambda)}{\sigma_t} = \frac{1}{1 + e^{2B/T}} \equiv \frac{1}{1 + e^{\varepsilon_\sigma(\lambda)}} \,, \tag{95}$$

$$n_\rho(k) = \frac{\rho_p(k)}{\rho_t} = \frac{2\cosh(\beta B)}{2\cosh(\beta B) + e^{\beta(\varepsilon(k) - h)}} \equiv \frac{1}{1 + e^{\varepsilon_\rho(k)}} \,. \tag{96}$$

Recall that $\beta = 1/T$. Notice that the last expression (96) is identical to (37). The relation between the total densities and particle densities can be written as

$$\begin{pmatrix} \rho_t \\ \sigma_t \end{pmatrix} = \begin{pmatrix} (2\pi)^{-1} \\ 0 \end{pmatrix} + \int\limits_{-\pi}^{\pi} \frac{dk}{2\pi} \begin{pmatrix} 0 & 0 \\ 1 & 0 \end{pmatrix} \begin{pmatrix} \rho_p(k) \\ \sigma_p(k) \end{pmatrix} \equiv d_0 + \hat{T} * \begin{pmatrix} \rho_p(k) \\ \sigma_p(k) \end{pmatrix} \,. \tag{97}$$

Here we have introduced the driving terms $d_0$ and the "scattering" kernel $\hat{T}$ which is determined by Bethe equation system (86),

$$\hat{T} = \begin{pmatrix} 0 & 0 \\ 1 & 0 \end{pmatrix} \,, \tag{98}$$

and the dressing is trivial in this case.

Denoting the vector of densities as $\vec{\rho}$ and introducing $n = \mathrm{diag}(n_\rho, n_\sigma)$, we see that $\vec{\rho}$ is a dressed version of $d_0$ in a sense that it is a solution of the following integral equation

$$\vec{\rho} = d_0 + \hat{T} n * \vec{\rho} \,. \tag{99}$$

Further, the magnetization (36) can be written as

$$\langle s_z(j, t) \rangle_T = \int\limits_{-\pi}^{\pi} h_\rho \rho_p(k) dk + \int\limits_{-\pi}^{\pi} h_\sigma \sigma_p(k) dk \,. \tag{100}$$

With $h_\rho = -1/2$ and $h_\sigma = 1$, being essentially the one-particle eigenvalues of the $S_z$. To address transport coefficients these quantities should be "dressed", either as via the magnetic dependence of quasienergies in the TBA solutions (95), (96), namely $h_{\rho,\sigma} = -\partial \varepsilon_{\rho,\sigma}(k)/\partial(2\beta B)$ or via the solution of the integral equation $h^{\mathrm{dr}} = h + \hat{T}^T n h^{\mathrm{dr}}$ (see for instance Eq. (3.22) in [13]), with $h^{\mathrm{dr}} = (h_\rho^{\mathrm{dr}}, h_\sigma^{\mathrm{dr}})^T$ and $h = (h_\rho, h_\sigma)^T$. The results read

$$h_\rho^{\mathrm{dr}} = -\frac{\tanh \beta B}{2} \,, \qquad h_\sigma^{\mathrm{dr}} = 1 \,. \tag{101}$$

The static covariance $\mathsf{C}$ computed within GHD for two types of particle ($\rho$ and $\sigma$) is given by [86]

$$\mathsf{C} = \int dk \rho_p(k)(1 - n_\rho(k))(h_\rho^{\mathrm{dr}})^2 + \int dk \sigma_p(k)(1 - n_\sigma(k))(h_\sigma^{\mathrm{dr}})^2 \,. \tag{102}$$

Using explicit formulas for the distribution functions (95), (96) and (101), we reproduce (69) obtained from the exact correlation function. The Drude weight can be computed in a similar manner

$$\mathsf{D} = \int dk \rho_p(k)(1 - n_\rho(k))(v_\rho^{\mathrm{eff}}(k) h_\rho^{\mathrm{dr}})^2 + \int dk \sigma_p(k)(1 - n_\sigma(k))(v_\sigma^{\mathrm{eff}}(k) h_\sigma^{\mathrm{dr}})^2 \,. \tag{103}$$

The effective velocities again can be computed from the quasienergies $v^{\text{eff}}_{\rho,\sigma} = -\beta^{-1}\partial_k \varepsilon_{\rho,\sigma}$, which gives $v^{\text{eff}}_\rho = \varepsilon'(k)$ and $v^{\text{eff}}_\sigma = 0$. Substituting all the quantities we recover the Drude weight (72).

Finally, for Onsager matrix $\mathfrak{L}$ can be computed as follows [12, 13]

$$\mathfrak{L} = \sum_{a,b=\rho,\sigma} \int \int \frac{dk_1 dk_2}{2} \rho_{p;a}(k_1)\rho_{p;b}(k_2)(1-n_a(k_1))(1-n_b(k_2))|v^{\text{eff}}_a(k_1)-v^{\text{eff}}_b(k_2)|$$

$$\times \left( \frac{T^{\text{dr}}_{b,a}(k_2,k_1)h^{\text{dr}}_b(k_2)}{\rho_{t;b}(k_2)} - \frac{T^{\text{dr}}_{a,b}(k_1,k_2)h^{\text{dr}}_a(k_1)}{\rho_{t;a}(k_1)} \right)^2 . \quad (104)$$

Here we assume that $\rho_{p;\rho} = \rho_p$ and $\rho_{p;\sigma} = \sigma_p$, and similarly for $\rho_{t;a}$. The dressed kernel can be understood again as a solution of integral equation similar to (99) $T^{\text{dr}} = \hat{T} + \hat{T}n * T^{\text{dr}}$, which has not effect due to nilpotency of the matrix $\hat{T}$ i.e. $T^{\text{dr}} = \hat{T}$. After this we arrive at the expression

$$\mathfrak{L} = \int_{-\pi}^{\pi} dk_1 \int_{-\pi}^{\pi} dk_2 \frac{\rho_p(k_1)}{\sigma_t^2(k_2)} \sigma_p(k_2)(1-n_\rho(k_1))(1-n_\sigma(k_2))|\varepsilon'(k_1)|, \quad (105)$$

which exactly reproduces Eq. (85).

# 6 Summary and Outlook

In this work we computed the key physical properties of spin transport in the t-0 model. Our computations are based on the exact presentation of the correlation functions in the thermodynamic limit in terms of the Fredholm determinants with their subsequent asymptotic analysis. This way we provide the first rigorous computation of spin diffusion for interacting quantum lattice systems. The results confirm the diffusion constant obtained earlier by semi-classical methods for infinite temperature [83], as well as the formula suggested by the generalized hydrodynamic [13].

In closely related deterministic models it was found that the fluctuations of spin transport are anomalous, even though the mean transport is still be diffusive [33, 66, 90–92]. It would be interesting to consider the full counting statistics also in the $t-0$ model, which is a fully quantum mechanical model.

In contrast to the diffusion found in our model it was found in [51] that in the Hubbard model the spin transport is superdiffusive at any finite coupling constant in zero magnetic field. This is not contradicting our results: in the Hubbard model the $U \to \infty$ limit is rather singular, and it can change the asymptotic behaviour of correlation functions.

It would be interesting to extend the present methods to spin diffusion in the folded XXZ model, which describes the infinite temperature dynamics of the XXZ model in the large anisotropy limit.

We hope to return to these questions in future work.

## Acknowledgements

We are thankful to Johannes Feldmeier, Sarang Gopalakrishnan, Jacopo De Nardis, Benjamin Doyon, Enej Ilievski, Miłosz Panfil, Romain Vasseur for useful discussions.

**Funding information** O.G. acknowledges support from the Polish National Agency for Academic Exchange (NAWA) through the Grant No. PPN/ULM/2020/1/00247.

## A Correlation functions of spinless fermions: Fredholm determinant representation

In this section we revisit derivation of the Fredholm determinant obtained in [84] with the "universal" use of the Wick's theorem according to [93]. We use finite lattice regularization, and perform minimal generalization of the string correlator (47) to consider the following correlation function of vertex operators

$$D_{\lambda\mu}(m,n;t) = \langle V_\mu^{(m)}(t)V_\lambda^{(n)}(0)\rangle, \tag{106}$$

where

$$V_\mu^{(m)}(0) \equiv V_\mu^{(m)} = \exp\left(i\mu \sum_{l=-L}^{m-1} c_l^+ c_l\right). \tag{107}$$

The lattice fermions are normalized as usual

$$\{c_m^+, c_n\} = \delta_{nm}. \tag{108}$$

The Fourier-transformed fermions $C_k$ defined as

$$C_k = \frac{1}{\sqrt{2L}} \sum_{m=-L}^{L} e^{-ikm} c_m, \qquad c_m = \frac{1}{\sqrt{2L}} \sum_k e^{ikm} C_k \tag{109}$$

makes the Hamiltonian diagonal $H = \sum_k \varepsilon(k) C_k^+ C_k$. Summation over momenta is taken over the Brillouin zone, meaning that

$$k = \frac{2\pi}{2L} n, \qquad n \in \mathbb{Z}, \qquad -\pi \le k < \pi. \tag{110}$$

For now, we assume that the average is computed over the vector that is given by

$$|\mathbf{q}\rangle \equiv |q_1 \dots q_n\rangle = C_{q_1}^+ \dots C_{q_n}^+ |0\rangle. \tag{111}$$

The vertex operator defined in (107) is a particular case of the group-like element $G(B)$ [93], which can be roughly defined as

$$G(B) =: e^{\sum_{k,p} C_p^+ B_{pk} C_k} :, \tag{112}$$

where the averaging is taken with respect to the mathematical vacuum $|0\rangle$. The matrix $B$ can be extracted from the action on the individual fermion

$$G(B)C_k^+ = \sum_p (\delta_{pk} + B_{pk}) C_p^+ G(B). \tag{113}$$

In fact, $G(B)$ can be defined via relation (113), which is valid also for the non-invertible group-like elements. The "group" property if reflected in the composition law

$$G(B')G(B) = G(B' + B + B'B), \tag{114}$$

which readily follows from (113). Finally, to evaluate (106) we will need the following corollary of Wick's theorem regarding the average of the group-like element on the state (111)

$$\langle \mathbf{q}|G(B)|\mathbf{q}\rangle = \det_{q \in \mathbf{q}, q' \in \mathbf{q}} (\delta_{qq'} + B_{qq'}). \tag{115}$$

Now let us compute the corresponding $B$ matrix for the vertex (107). Commuting it with the fermion creation operator, we obtain

$$V_{\mu}^{(m)} c_a^+ = \left[1 + (e^{i\mu} - 1)\theta(a < m)\right] c_a^+ V_{\mu}^{(m)}, \tag{116}$$

or for the Fourier modes

$$V_{\mu}^{(m)} C_k^+ = \sum_p (\delta_{pk} + B_{pk}) C_p^+ V_{\mu}^{(m)}, \tag{117}$$

where

$$[B_{\mu}^{(m)}]_{pk} = \frac{e^{i\mu} - 1}{2L} \frac{e^{im(k-p)} - e^{-iL(k-p)}}{e^{i(k-p)} - 1}, \tag{118}$$

for $p \neq k$, while diagonal components are given by

$$[B_{\mu}^{(m)}]_{kk} = \frac{e^{i\mu} - 1}{2L}(L + m). \tag{119}$$

Note that if we keep $L$ dependence explicitly then the diagonal part comes as l'Hopital rule.

Time dependence can be easily included as well

$$[B_{\mu}^{(m)}]_{pk}(t) = [B_{\mu}^{(m)}]_{pk} e^{it(\varepsilon(p) - \varepsilon(k))}. \tag{120}$$

Now employing (114) and (115) we arrive at

$$D_{\lambda\mu}(m, n; t) = \det \mathcal{A}, \tag{121}$$

with

$$\mathcal{A}_{ij} = \delta_{q_i q_j} + [B_{\mu}^{(m)}]_{q_i q_j} e^{i(\varepsilon(q_i) - \varepsilon(q_j))t/2} + [B_{\lambda}^{(n)}]_{q_i q_j} e^{i(\varepsilon(q_i) - \varepsilon(q_j))t/2} \\ + e^{i\varepsilon(q_i)t/2} \sum_k [B_{\mu}^{(m)}]_{q_i k} e^{-i\varepsilon(q_k)t} [B_{\lambda}^{(n)}]_{k q_j} e^{i\varepsilon(q_j)t/2}. \tag{122}$$

Let us evaluate the sum in this expression treating $L$ as a large parameter. First, we rewrite the sum identically

$$\sum_k [B_{\mu}^{(m)}]_{q_i k} e^{-i\varepsilon(q_k)t} [B_{\lambda}^{(n)}]_{k q_j} = \frac{(e^{i\lambda} - 1)(e^{i\mu} - 1)}{(2L)^2} \sum_k \frac{\Theta(q_i, q_j; k)}{(e^{ik} - e^{iq_i})(e^{iq_j} - e^{ik})} e^{-it\varepsilon(q_k) + iq_i + ik}, \tag{123}$$

with

$$\Theta(q_i, q_j; k) = e^{i(q_j n - q_i m + k(m-n))} + e^{iL(q_i - q_j)} - e^{i(n q_j + L q_i - (L+n)k)} - e^{i((L+m)k - m q_i - L q_j)}. \tag{124}$$

For $q_i \neq q_j$ we present this expression as

$$\sum_k [B_{\mu}^{(m)}]_{q_i k} e^{-i\varepsilon(k)t} [B_{\lambda}^{(n)}]_{k q_j} = \frac{(e^{i\lambda} - 1)(e^{i\mu} - 1)}{2L} \frac{V - W}{e^{iq_i} - e^{iq_j}}, \tag{125}$$

with

$$V = \frac{L + m}{2L}(e^{iL(q_i - q_j)} - e^{in(q_j - q_i)})e^{-it\varepsilon(q_i) + iq_i} + \frac{1}{2L} \sum_{k \neq q_i} \frac{\Theta(q_i, q_j; k)}{e^{iq_i} - e^{ik}} e^{-it\varepsilon(k) + iq_i + ik}, \tag{126}$$

$$W = \frac{L+n}{2L}(e^{im(q_j-q_i)} - e^{iL(q_i-q_j)})e^{-it\varepsilon(q_j)+iq_i} + \frac{1}{2L}\sum_{k\neq q_j}\frac{\Theta(q_i,q_j;k)}{e^{iq_j}-e^{ik}}e^{-it\varepsilon(k)+iq_i+ik}. \tag{127}$$

After these preparations let us evaluate the limit of these sums as Riemann integral. The highly oscillating terms can be thrown away and the corresponding superficial divergences should be treated as principal value (as we demonstrate in Appendix (A.1))

$$\frac{1}{2L}\sum_{k\neq q}\frac{e^{iL(k-q)}-1}{e^{iq}-e^{ik}}f_k = -\frac{\text{v.p.}}{2\pi}\int_{-\pi}^{\pi}dk\frac{f_k}{e^{iq}-e^{ik}} = -\frac{1}{2\pi}\int_{-\pi}^{\pi}dk\frac{f_k-f_q}{e^{iq}-e^{ik}}. \tag{128}$$

For $q_i = q_j$ we can formally compute l'Hopital's limit to obtain

$$\mathcal{A}_{ii} = \frac{1+e^{i\lambda+i\mu}}{2} + O(1/L), \tag{129}$$

which means that

$$\det\mathcal{A} \sim (\cos(\lambda+\mu))^{2L}e^{iL(\lambda+\mu)}. \tag{130}$$

So, we have to demand $\mu = -\lambda$ to obtain non-zero answer as $L \to \infty$. Once this condition is assumed we can get rid of terms proportional to $e^{i(q_i-q_j)L}$ in all expressions in Eq. (122), similarly to (128). Further assuming that $m, n \ll L$, we present

$$V = -\frac{e^{-it\varepsilon(q_i)+iq_i+in(q_j-q_i)}}{2} - e^{inq_j-i(m-1)q_i}\hat{E}(q_i), \tag{131}$$

$$W = \frac{e^{im(q_j-q_i)-it\varepsilon(q_j)+iq_i}}{2} - e^{inq_j-i(m-1)q_i}\hat{E}(q_j), \tag{132}$$

with

$$\hat{E}(q) = \frac{\text{v.p.}}{2\pi}\int_{-\pi}^{\pi}dk\frac{e^{i(m-n+1)k-it\varepsilon(k)}}{e^{iq}-e^{ik}}, \tag{133}$$

and finally

$$\mathcal{A}_{ij} = -\frac{|e^{i\lambda}-1|^2}{2L}\frac{\hat{E}(q_i)-\hat{E}(q_j)}{e^{iq_i}-e^{iq_j}}e^{inq_j-i(m-1)q_i+i(\varepsilon(q_i)+\varepsilon(q_j))t/2}$$
$$-i\sin(\lambda)\frac{e^{im(q_j-q_i)+iq_i+i(\varepsilon(q_i)-\varepsilon(q_j))t/2} - e^{i(\varepsilon(q_j)-\varepsilon(q_i))t/2+iq_i+in(q_j-q_i)}}{2L(e^{iq_i}-e^{iq_j})}. \tag{134}$$

To literally reproduce results of [84] we would need the following relation

$$\frac{e^{iq_i}}{e^{iq_i}-e^{iq_j}} = \frac{1}{2}\frac{e^{iq_i}+e^{iq_j}}{e^{iq_i}-e^{iq_j}} + \frac{1}{2} = \frac{1}{2i\tan((q_i-q_j)/2)} + \frac{1}{2}. \tag{135}$$

Moreover, using this relation we can connect $\hat{E}(q)$ with $E(m-n,q)$ defined in (54), we have

$$\hat{E}(q) = -\frac{E(m-n,q)}{2i} + \frac{G(m-n)}{2}, \tag{136}$$

where $G(x)$ is defined as

$$G(x) = \int_{-\pi}^{\pi}\frac{dq}{2\pi}e^{-it\varepsilon(q)+ixq} = i^x J_x(2t). \tag{137}$$

Since $G(x)$ does not depend on $q$ it does not contribute to matrix elements $\mathcal{A}_{ij}$ (134). Finally, assuming $x = m - n$ and using notations (52) and (53) we obtain

$$\mathcal{A}_{ij} = e^{-i(m+n+1)q_i/2}\left(\delta_{ij} + \frac{1}{2L}\frac{E_+(x,q_i)E_-(x,q_j) - E_+(x,q_j)E_-(x,q_i)}{\sin\frac{q_i-q_j}{2}}\right)e^{i(m+n+1)q_j/2}. \quad (138)$$

The conjugation factors will cancel in the determinant $\det\mathcal{A}$. Further, taking into account the level spacing (110) and introducing density of states $n_\rho(k)$ in $L \to \infty$ limit we recover (50). For the formal proof of the validity of injection of the density distribution after averaging over the thermal ensemble see, for instance, appendix A in [94].

## A.1  Proof of the lemma

Here we present some comments on transformation of the sum (128) into integrals. First we notice the following identity

$$\frac{1}{2L}\sum_k = \frac{1}{2\pi}\oint_C \frac{dk}{e^{2ikL} - 1}, \quad (139)$$

where counterclockwise contour $C$ encircles solution of $e^{2ikL} - 1 = 0$ that are inside the first Brillouin zone ($-\pi < k \le \pi$). For summation of the smoothing varying function on these interval we can present $C$ as the combination of the contours above and below the real axis

$$C = \gamma_1 \cup \gamma_2, \quad (140)$$

$$\gamma_1 = \{k + i\epsilon | k \in [\pi, -\pi]\}, \qquad \gamma_2 = \{k - i\epsilon | k \in [-\pi, \pi]\}, \quad (141)$$

where $\epsilon \ll 1 \ll L\epsilon$. This way we may ignore contribution from the contour $\gamma_2$ while contribution from $\gamma_1$ actually gives normal Riemann integral

$$\frac{1}{2L}\sum_k f_k = \frac{1}{2\pi}\int_{\gamma_1} dk\frac{f_k}{-1} = \int_{-\pi}^{\pi}\frac{dk}{2\pi}f_k. \quad (142)$$

Let us consider

$$S_q \equiv \frac{1}{2L}\sum_{k\ne q}\frac{e^{iL(k-q)} - 1}{e^{iq} - e^{ik}}f_k = \frac{1}{2\pi}\oint_{C_q}\frac{dk}{e^{2ikL} - 1}\frac{e^{iL(k-q)} - 1}{e^{iq} - e^{ik}}f_k, \quad (143)$$

where in $C_q$ we emphasize that point $k = q$ is not encircled. Taking into account that $e^{2iqL} = 1$ we can present

$$S_q = \frac{1}{2\pi}\oint_{C_q}\frac{dk}{e^{2i(k-q)L} - 1}\frac{e^{iL(k-q)} - 1}{e^{iq} - e^{ik}}f_k = \frac{1}{2\pi}\oint_{C_q}\frac{dk}{e^{i(k-q)L} + 1}\frac{f_k}{e^{iq} - e^{ik}}, \quad (144)$$

or including residue, and transforming as in the regular case we get

$$S_q = \frac{1}{2\pi}\oint_C\frac{dk}{e^{i(k-q)L} + 1}\frac{f_k}{e^{iq} - e^{ik}} + \frac{e^{-iq}}{2} = -\frac{1}{2\pi}\int_{-\pi+i\epsilon}^{\pi+i\epsilon} dk\frac{f_k}{e^{iq} - e^{ik}} + \frac{e^{-iq}}{2}. \quad (145)$$

Transforming further we can present

$$S_q = -\frac{1}{2\pi}\int_{-\pi}^{\pi} dk\frac{f_k}{q - k - i\epsilon}\frac{q - k}{e^{iq} - e^{ik}} + \frac{e^{-iq}}{2} = -\frac{\text{v.p.}}{2\pi}\int_{-\pi}^{\pi} dk\frac{f_k}{e^{iq} - e^{ik}}, \quad (146)$$

which basically means that you can throw away $e^{ikL}$ from the integration if you treat everything in a primary value sense. At the final step we use the following identity

$$\frac{\text{v.p.}}{2\pi}\int_{-\pi}^{\pi} dk\frac{1}{e^{iq} - e^{ik}} = 0, \quad (147)$$

leading to Eq. (128).

# B  Series expansion

Let us expand the string correlator $\mathcal{F}_\lambda(x,t)$ in (49) at $\lambda = 0$. Taking into account the following expansion of the determinant

$$\det(1+R) = 1 + \text{Tr}R + \frac{(\text{Tr}R)^2 - \text{Tr}R^2}{2} + O(R^3), \tag{148}$$

we obtain

$$\mathcal{F}_\lambda(x,t) = 1 - i\lambda \int_{-\pi}^{\pi} \frac{dk}{2\pi} n_\rho(k)(t\varepsilon'(k) - x)$$

$$+ \frac{\lambda^2}{2} \int_{-\pi}^{\pi} \frac{dk}{2\pi} n_\rho(k) e^{it\varepsilon(k)} \partial_k [e^{-ikx} E(k)] + i\frac{\lambda^2}{2} \int_{-\pi}^{\pi} \frac{dk}{2\pi} n_\rho(k) x e^{it\varepsilon(k) - ikx} E(k)$$

$$+ \frac{\lambda^2}{2} \int_{-\pi}^{\pi} \frac{dk}{2\pi} \int_{-\pi}^{\pi} \frac{dq}{2\pi} n_\rho(k) n_\rho(q) \left( \frac{\sin\left[\frac{t}{2}(\varepsilon(k) - \varepsilon(q)) - \frac{x(k-q)}{2}\right]}{\sin\frac{k-q}{2}} \right)^2$$

$$- \frac{\lambda^2}{2} \int_{-\pi}^{\pi} \frac{dk}{2\pi} n_\rho(k)(t\varepsilon'(k) - x) \int_{-\pi}^{\pi} \frac{dq}{2\pi} n_\rho(q)(t\varepsilon'(q) - x). \tag{149}$$

This way,

$$\hat{S}\mathcal{F}_\lambda(x,t) = \lambda^2 \int_{-\pi}^{\pi} \frac{dk}{2\pi} n_\rho(k) \int_{-\pi}^{\pi} \frac{dq}{2\pi} n_\rho(q)$$

$$+ \lambda^2 \int_{-\pi}^{\pi} \frac{dk}{2\pi} n_\rho(k) \int_{-\pi}^{\pi} \frac{dq}{2\pi} e^{it(\varepsilon(k) - \varepsilon(q)) - ix(k-q)}(1 - n_\rho(q)) + O(\lambda^3). \tag{150}$$

# C  Kernels

In this chapter we compare our answers with those in Ref. [59]. To do so we have to introduce one more kernel

$$\mathcal{Q}(x,\lambda|k,q) = \frac{\ell_+(x,k)\ell_-(x,q) - \ell_-(x,k)\ell_+(x,q)}{2\pi \tan\frac{k-q}{2}} - \frac{1}{2}\frac{1 - \cos\lambda}{2\pi} G(x)\ell_-(x,k)\ell_-(x,q), \tag{151}$$

where

$$G(x) = \int_{-\pi}^{\pi} \frac{dq}{2\pi} e^{-it\varepsilon(q) + ixq} = i^x J_x(2t), \tag{152}$$

with $\varepsilon(q) = -2\cos(q)$. Further one can notice that

$$E(x+1,k) = e^{ik}E(x,k) + ie^{ik}G(x) + iG(x+1), \tag{153}$$

or

$$E(x-1,k) = e^{-ik}E(x,k) - ie^{-ik}G(x) - iG(x-1). \tag{154}$$

This leads to

$$\ell_+(x+1,k) = e^{ik/2}\ell_+(x,k) + \sqrt{\rho(k)}\frac{1-\cos\lambda}{2}iE_-(x,k)\left(e^{ik/2}G(x)+e^{-ik/2}G(x+1)\right), \quad (155)$$

or

$$\ell_+(x+1,k) = e^{ik/2}\ell_+(x,k) + \frac{1-\cos\lambda}{2}i\ell_-(x,k)e^{ik/2}G(x) + \frac{1-\cos\lambda}{2}i\ell_-(x+1,k)G(x+1). \quad (156)$$

This way

$$\mathcal{U}(x+1,\lambda|k,q) = \mathcal{Q}(x,\lambda|k,q)$$
$$+ \frac{i\ell_+(x,k)\ell_-(x,q)+i\ell_+(x,q)\ell_-(x,k)}{2\pi} - \frac{1}{2}\frac{1-\cos\lambda}{2\pi}G(x)\ell_-(x,k)\ell_-(x,q), \quad (157)$$

$$\mathcal{U}(x-1,\lambda|k,q) = \mathcal{Q}(x,\lambda|k,q)$$
$$- \frac{i\ell_+(x,k)\ell_-(x,q)+i\ell_+(x,q)\ell_-(x,k)}{2\pi} - \frac{1}{2}\frac{1-\cos\lambda}{2\pi}G(x)\ell_-(x,k)\ell_-(x,q). \quad (158)$$

Additionally, we can present

$$e^{i(k-q)/2}\mathcal{U}(x,\lambda|k,q) = \mathcal{Q}(x,\lambda|k,q) + \frac{i\ell_+(x,k)\ell_-(x,q)-i\ell_+(x,q)\ell_-(x,k)}{2\pi}$$
$$+ \frac{1}{2}\frac{1-\cos\lambda}{2\pi}G(x)\ell_-(x,k)\ell_-(x,q). \quad (159)$$

Let us introduce three rank -one operators

$$R_1(k,q) = \frac{i}{2\pi}\left(\frac{1}{1+\mathcal{Q}}l_+\right)(k)\left(\frac{1}{1+\mathcal{Q}}l_-\right)^T(q), \quad (160)$$

$$R_2(k,q) = \frac{i}{2\pi}\left(\frac{1}{1+\mathcal{Q}}l_-\right)(k)\left(\frac{1}{1+\mathcal{Q}}l_+\right)^T(q), \quad (161)$$

$$R_3(k,q) = \frac{1-\cos\lambda}{4\pi}G(x)\left(\frac{1}{1+\mathcal{Q}}l_-\right)(k)\left(\frac{1}{1+\mathcal{Q}}l_-\right)^T(q). \quad (162)$$

Than taking into account that

$$\det(1+e^{i(k-q)/2}\mathcal{U}(x,\lambda|k,q)) = \det(1+\mathcal{U}(x,\lambda|k,q)) = \mathcal{D}(x,t). \quad (163)$$

We obtain

$$\frac{\mathcal{D}(x+1,t)}{\det(1+\mathcal{Q})} = \det(1+R_1+R_2-R_3), \quad (164)$$

$$\frac{\mathcal{D}(x-1,t)}{\det(1+\mathcal{Q})} = \det(1-R_1-R_2-R_3), \quad (165)$$

$$\frac{\mathcal{D}(x,t)}{\det(1+\mathcal{Q})} = \det(1-R_1+R_2+R_3). \quad (166)$$

Further, taking into account that linear combination of $R_1$ ($R_2$) and $R_3$ is a rank-one operator, we obtain

$$\frac{\mathcal{D}(x+1,t)}{\det(1+\mathcal{Q})} = 1 + \text{Tr}(R_1+R_2-R_3) + \text{Tr}(R_1-R_3)\text{Tr}R_2 - \text{Tr}(R_1-R_3)R_2, \quad (167)$$

$$\frac{\mathcal{D}(x-1,t)}{\det(1+\mathcal{Q})} = 1 - \mathrm{Tr}(R_1 + R_2 + R_3) + \mathrm{Tr}(R_1 + R_3)\mathrm{Tr}R_2 - \mathrm{Tr}(R_1 + R_3)R_2\,, \tag{168}$$

$$\frac{\mathcal{D}(x,t)}{\det(1+\mathcal{Q})} = 1 + \mathrm{Tr}(R_2 + R_3 - R_1) + \mathrm{Tr}(R_3 - R_1)\mathrm{Tr}R_2 - \mathrm{Tr}(R_3 - R_1)R_2\,. \tag{169}$$

This way,

$$\frac{\mathcal{D}(x+1,t) + \mathcal{D}(x-1,t) + 2\mathcal{D}(x,t)}{\det(1+\mathcal{Q})} = 4\,. \tag{170}$$

Or in other words

$$\det(1 + \hat{\mathcal{U}}(\lambda)) - \det(1 + \mathcal{Q}(\lambda)) = \frac{2\mathcal{D}(x,t) - \mathcal{D}(x-1,t) - \mathcal{D}(x+1,t)}{4}\,. \tag{171}$$

This statement is enough to prove the equivalence of our results to those in [59].

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
