# Peer review of "Finite temperature spin diffusion in the Hubbard model in the strong coupling limit"

_SciPost Physics, doi:SciPost Phys. 15, 073 (2023)_

## Round 2 · Referee Report · Anonymous (Referee 1) · 2023-5-13

Strengths

1-interesting model
2-actual interest
3-exact calculation
4-well written

Weaknesses

1-missing comparison with other calculations
2-missing physical information
3-what means "absence of bias" in the abstract ?

Report

The authors present an evaluation of the spin-spin correlation function at finite temperature in the infinite-U 1D Hubbard model. The exact result is in the form of Fredholm determinants
that must in the end be evaluated numerically.
They give long time asymptotic expressions from which they extract the Drude weight and diffusion constant.
The compare their results with the those obtained from the GHD approach.

I think it would be useful to also compare them with results from Mazur bound in the high temperature limit which are transparent and very easy to obtain.

Although it is a rather mathematical study, it would be very interesting to present and discuss physical results on the Drude weight and diffusion constant e.g. as a function of temperature, magnetic field (magnetization).
After these additions/discussion I recommend publication in this journal.

---

## Round 2 · Referee Report · Anonymous (Referee 2) · 2023-5-29

Strengths

  • very impressive exact calculations of a quasi-interacting model
  • first explicit check of a spin diffusion constants in integrable systems by means of wave function expression

Weaknesses

  • not many just minor points

Report

This is the first check of the expression for the Onsager matrix obtained in https://journals.aps.org/prl/abstract/10.1103/PhysRevLett.121.160603 directly via the miscroscopic wave function in an interacting integrable model. The calculations are clean and well presented, definitely a work to be published in SciPost. Here minor comments:

  • first, it would be better to stress that eq 103 has been derived first and only in the following paper, https://journals.aps.org/prl/abstract/10.1103/PhysRevLett.121.160603 , with this one as its long version https://scipost.org/10.21468/SciPostPhys.6.4.049. There is a common misunderstanding stated here in the introduction: GHD is not a tool to compute diffusion constants (but rather a theory that needs the input of the exact form of the diffusion constants). The latter has been computed first using a form factor expansion (in the two references above), then later also by hydrodynamic projection and kinetic theory, never by "GHD".
  • Can the author compute integrated current-current correlator also, in order to obtain the Onsager matrix?
  • Can the authors give an analytical expression for the "diffusion constant" at finite time, i.e. figure 1 inset? It would be a nice addition.
  • Why is rho = 2/3 at infinite temperature?
  • I cannot see where the scattering T in eq 97 is derived
  • Could the author's comment on the 1/U correction ? Spin diffusion constant is indeed expected to be infinite at finite U and zero magnetisation.
  • Section 5 i would rather call it "Thermodynamic of the model and GHD diffusion constant". It would also be nice to make clear how to derive the TBA of the model from the large U limit of the TBA of the Hubbard model.

Requested changes

see report

---

## Round 3 · Author Response

We made a couple of changes, and please find our replies below.

---

## Round 3 · List of Changes

Reply to Referee 1
-We added a new figure, namely Fig. 2, which shows the temperature and magnetic field dependence of the spin Drude weight and the spin diffusion constant.
-The Drude weight for the spin transport is actually a simple free fermionic result, multiplied by a factor which depends on the magnetic field. Therefore, we did not add a discussion of the Mazur bounds. The most important result of our paper is in our point of view the spin diffusion constant, for which there have been no exact results yet. Therefore, we did not intend to discuss the multiple approaches for the ballistic part of the spin transport, in those cases when this part is non-zero. We hope the referee can agree with this choice.
-In the abstract we replaced the expression "absence of bias" with "absence of magnetic field".
Reply to Referee 2
-We have added the required citations to eq. (103). Also, we modified the text in the Introduction, where diffusion and GHD are mentioned. Let us add here, that it is not clear to us, what does "GHD" actually encompass. Perhaps those authors who worked on the theory from the beginning have a different view about this, than other researchers. To us the umbrella term "GHD" would include both the thermodynamic form factors, and hydro projections. In the current version we added a mention of both concrete methods to the introduction, whereas keeping the expression "GHD framework". We hope this is acceptable to the referee...
-In the paper we already have the specific components of the Onsager matrix, see eq. (85).
But in principle, this matrix can be computed directly from the current-current correlator.
-We can extract the finite time diffusion constant only numerically. We added a plot about it.
-Infinite temperature means here that the three local states have equal probability. One state is the vacuum, other two states are electron states with spin up and down. Therefore, the density is 2/3. Of course, if we think about the original Hubbard model, this picture applies only if T >> 1 but T << U. Because if the temperature is larger than the original coupling constant U, then the doublon degrees of freedom are also occupied. Right now we did not add this explanation to the text, but if the referee requires, we could add it.
-We added eq. (98) which shows the scattering kernel, derived directly from the Bethe equations.
-Unfortunately, we can not add anything about 1/U corrections. The limits U->infty and t->infty do not commute. This is why at any finite U the diffusion constant is infinite. We already commented about this in the
Conclusions. Unfortunately, we can not add more. The model is singular, and perhaps this is not very physical. Nevertheless, this is still the first quantum spin chain model, where the diffusion constant was calculated, so this gives a justification for the work we did.
-We replaced the title of that Section.

---

## Editorial Decision

published